# A casino in my pocket: Gratifications associated with obsessive and harmonious passion for mobile gambling

**Eoin Whelan**[1]*, **Samuli Laato**[2], **A. K. M. Najmul Islam**[2], **Joël Billieux**[3,4]

**1** Business Information Systems, J.E. Cairnes School of Business and Economics, NUI Galway, Ireland,
**2** Department of Future Technologies, University of Turku, Turku, Finland, **3** Institute of Psychology,
University of Lausanne, Lausanne, Switzerland, **4** Centre for Excessive Gambling, Lausanne University
Hospitals (CHUV), Lausanne, Switzerland

* eoin.whelan@nuigalway.ie

CANADA

**Data Availability Statement:** The data used in this
study is held in the OSF public repository https://
osf.io/643ja/.

**Funding:** The authors received no specific funding
for this work.

## Abstract

Mobile gambling differs from land-based and traditional forms of gambling in that the opportunity to place bets and engage with casinos is constantly present and easily accessible. Instead of going to a physical bookmaker or casino, mobile gambling is done quickly and swiftly, anytime, anywhere, with a few taps on a mobile device. Previous studies reveal mobile gambling has managed to reach new audiences especially amongst younger people. Gambling harms can have severe adverse effects on individuals, families and society. However, for a subgroup of highly involved individuals, gambling can be considered a harmonious passion that permits frequent gambling without elevating individual's risks of experience problem gambling manifestations. Combining the Uses and Gratifications (U&G) and Dualistic Model of Passion (DMP) frameworks, the present study aims to determine if and how the different gratifications sought from mobile gambling are susceptible to explaining non-problematic versus problematic patterns in highly involved gamblers. Data were collected over two waves from a global sample of mobile gamblers (N = 327). Results emphasize that the motivational underpinnings of mobile gambling (as measured by the U&G) differ in obsessive versus harmonious passion. Obsessive passion is associated with poor mood and problematic gambling. In contrast, harmonious passion for mobile gambling is associated with positive mood but is unrelated to problematic gambling. Based on these findings, and given that problematic gambling is an internationally relevant public health issue (the prevalence of problem gambling is estimated to range from 0.1% to 5.8% in different countries), we suggest interventions focusing on specific uses and gratifications associated with an obsessive passion for mobile gambling may be effective in reducing problematic usage patterns.

## Introduction

Gambling is a popular and common pastime across many countries, yet the harms associated to gambling are nowadays recognized as an important public health issue [1]. For a subgroup

**Competing interests:** The authors have declared that no competing interests exist.

of vulnerable individuals, gambling involvement can be pathological and reflects a mental disorder. Individuals with gambling disorder are characterized by the inability to control their gambling behavior, which is ultimately susceptible to translate into severe functional impairment such as financial crisis or familial disruption [2]. The prevalence of problem gambling is estimated to vary between 0.12% and 5.8% in different countries around the globe [3]. A wide range of social (e.g., peer influence), environmental (e.g., negative life event), psychological (e.g., poor emotion regulation skills), and neurobiological factors (e.g., development of incentive salience) have been shown to promote problematic gambling patterns [4,5]. For most people though, gambling is a non-problematic recreational activity. In fact, some people gamble frequently with a sense of volition, and without experiencing negative consequences such as financial problems [6–8].

The gambling landscape has been subjected to an unprecedented evolution since the apparition of online and mobile gambling opportunities [9,10]. The present study is concerned with mobile gambling, whereby people gamble online using their smartphones through specially designed applications and websites [11]. Mobile technology introduces new dimensions and opportunities to gamble. A recent literature review concludes that mobile gambling has distinctive features compared to more traditional forms of gambling [12]. First and foremost, mobile devices enable gambling at will, ubiquitously and independent of location [13], with mobile app use suggested to promote a form of gambling that is more impulsive and habitual in nature [14].

Through an array of data collection opportunities provided by smartphones, mobile gambling apps are able to personalize gambling, in-play promotions for example, in a way traditional or even other forms of online gambling cannot. Gambling apps can be run on the background, prompting notifications of gambling opportunities. They can technically utilize data such as the users' location, list of contacts, credit card balance, and use history among other things, to which even the most sophisticated modern slot machines do not have access to. In many instances, the types of games played are different when the medium is a mobile device [12]. Gambling through an app also removes any stigma associated with placing a bet directly with another person, which has been linked to an increase in sports gambling among young men [10]. Given these new experiences and opportunities afforded by mobile technology, there is a public health need to better understand the psychology associated with problematic versus non-problematic patterns of mobile gambling behaviors.

Previous research concludes that mobile gamblers are at increased risk of developing harmful gambling habits in comparison to land-based and other non-mobile gambling types [11]. A recent study capitalizing on behavioral tracking data also reports that sport bettors using mobile apps are more likely to present hazardous patterns of gambling [15]. Given the growing data suggesting that mobile gambling is riskier for individuals than other types of gambling activities, research is needed to identify individual factors that might constitute specific risk and protective factors towards problematic mobile gambling behaviors. Although a recent stream of research has focused on the concept of healthy passion to emphasize the potentially adaptive and functional features of gambling involvement [6,8,16–18], uncertainty still abounds regarding the psychological dimensions distinguishing non-problematic versus problematic intensive involvement in mobile gambling. Past research suggests that gambling motives play a pivotal role when it comes to account for problematic versus non-problematic gambling involvement [19–21]. For example, it has been shown that enhancement-related motives (e.g., feeling high) and coping-related motives (e.g., to relieve negative affect) are vulnerability factors for problematic gambling, which is not the case of social motives (e.g., meeting and sharing with people in the gambling context) [22]. In this context, our objective is here to determine how the antecedents and motivational underpinning of both problematic and

non-problematic intensive involvement in mobile gambling. To do so, we will capitalize on both the uses and gratification (U&G) theory from the communication and media sciences [23] and the dualistic model of passion (DMP) theory from the psychological sciences [24].

The U&G approach is a dominant theoretical lens in the study of the uses and impacts of particular media, from radio and TV to social media and online gaming [25]. U&G research has its foundations the structural–functionalist systems approach to discerning the interaction between living entities and their context [26]. Diverging from other media consumption theories, U&G identifies the needs and desires individuals have to use a particular media [27]. Choices as to what level of engagement the user has with media are made based upon the perceived psychological needs that the content of a media is susceptible to fulfil [23]. Thus, U&G posits that media consumption behavior is freely guided by psychological gratifications derived from using the product as well as practical reasons to use it. Additionally, the effects of the individual's media use can be largely explained by understanding the individual's purpose for using the media [26].

To the best of our knowledge, no study to date has applied the U&G perspective to understand why people engage with mobile gambling. However, pertinent to the current study, a number of recent investigations do shed light on the U&G associated with mobile games. In this context, *social interaction* has been found to be one of the most prominent U&G [28–30], contributing to in-app purchase intention in mobile games [28] and platform loyalty [31]. *Information seeking* drives the use of media such as TV [32] and the Internet [33], an association which has also found to be relevant in mobile gaming [30], and could constitute a focal motivation for seeking out the best gambling options. While immersed in a mobile game, users become distracted from other stressful aspects of their lives. Seeking the gratification of *relaxation* predicts engagement with mobile games [30,34]. It is worth noting that adaptive *relaxation* has to be distinguished from maladaptive forms of escapism (i.e. coping with adverse emotions, negative life events, or psychopathological symptoms) which have been systematically linked with problematic gaming [35,36]. For many people today, their smartphone is often the first port of call when boredom arises [37]. Exceptionally successful smartphone games such as Candy Crush, Angry Birds, and Pokémon Go are downloaded to not only provide the gratification of *passing time* [38] but also to *entertain* [34,39]. Finally, *arousing emotions* is also a well-documented gratification linked to the increased length of time people engage with video games [26], a motive which extends to mobile apps [40,41]. While the six U&G identified above are influential in explaining engagement and outcomes from mobile games, our purpose here is to determine if these U&G triggers are different for people presenting high but unproblematic involvement versus problematic involvement in mobile gambling.

The DMP [42,43] is relevant in demarcating high versus pathological involvement in specific leisure activities. According to this framework, passion is defined as a *"strong inclination toward a specific object, activity, concept or person that one loves (or at least strongly likes) and highly values, that is part of their identity, and that leads one to invest time and energy in the activity on a regular basis"* [26; p. 42]. For many people, holding a passion for some activity, be it sport, music, games, writing, work, or family, is what makes life meaningful and worth living [44]. Passions, however, can be a doubled edged sword. They can be functional or dysfunctional, depending on how the activity is internalized into the individual's core identity and self. The DMP captures this duality by conceptualizing passion as consisting of two related but conceptually distinct components; "harmonious" and "obsessive" passion [43].

*Harmonious passion* refers to the motivation to feel free to engage in a given activity considered important, even if it takes up space in the person's life, is not in conflict with other domains of their life and the person still remains in control. Inversely, *obsessive passion* is characterized by an irrepressible pressure to participate in the activity, which results in interfering

with other domains in the person's life and eventually involves functional impairment [24,43]. Both obsessive passion and harmonious passion are associated with the desire to continue participating and feeling joy during the activity [43]. In this respect, the DMP would suggest that people with harmonious passion for mobile gambling still remain in control of the activity, have intention to continually use mobile gambling, live positive outcomes, and would be less likely to experience disordered gambling symptoms. In contrast, people with obsessive mobile gambling passions would have intention to continually use mobile gambling, live negative outcomes, and would be more likely to have gambling problems. Indeed, a recent review of studies having applied the DMP to gambling concludes that a harmonious passion for gambling is generally associated with positive outcomes, such as vitality, while an obsessive passion generally results in problems like negative emotions, psychopathological symptoms, and gambling disorder [7].

The DMP constructs of harmony and obsession present similarities with constructs included in influential addiction models such as the classic dichotomy between "liking" and "wanting" (operationalized in the incentive-sensitization theory [45]) or the distinction between "controlled use" and "impulsive use" (operationalized in dual-process models, see e.g. [46]. Yet, an important difference between the DMP and these models is that the latter essentially focus on neurobiological processes whereas the DMP is conceptualized within a broader psychological framework that consider a wider range of psychological processes (including self, identity, or personality-related processes). Although the DMP has only rarely been used in the context of drug use, a study focusing on recreational marijuana consumption showed that harmonious passion toward its consumption is related to elevated life satisfaction and lower consumption-related negative consequences, whereas the opposite pattern is observed with regard to obsessive passion scores [47].

In this study, we capitalized on both the U&G and DMP framework in order to identify the different media gratification triggers which may explain why the opposing passions and consequences (problematic gambling, general mood) of mobile gambling manifest. As this study is exploratory in nature, we do not present specific hypotheses to be tested, rather a research model from which the implications of our study are interpreted. This research model is illustrated in Fig 1.

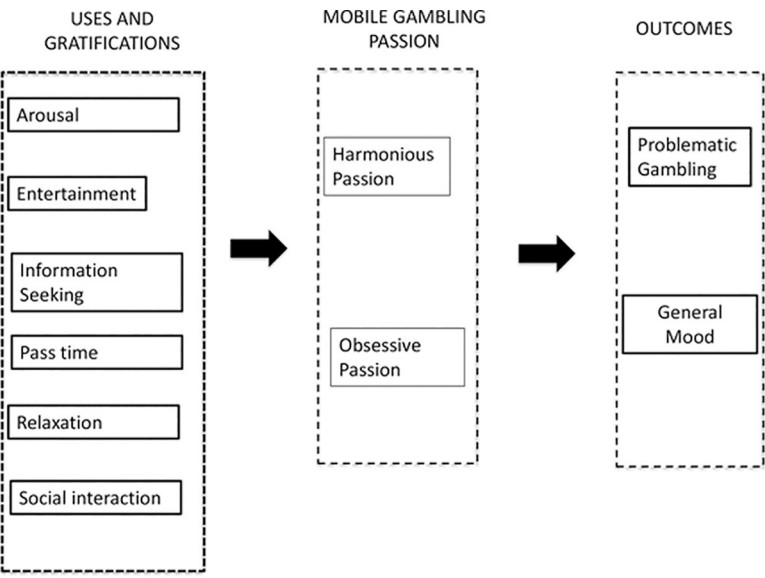

**Fig 1. The research model.**

## Method

### Data collection and procedure

The present study is an ancillary project drawn from a larger dataset; other results from this dataset will be presented elsewhere. Survey data were collected over two points in time through Amazon's Mechanical Turk (MTurk) platform. An initial pilot test of the survey with 92 respondents confirmed the soundness of the implemented scales, and also resulted in the rewording of the participant instructions and a small number of items to improve clarity. The time lagged design separated predictor variables from outcomes which reduces common method bias [48] and lends stronger support for causal arguments [49]. The U&G and DMP variables were collected at t1. One week later, outcome variables were gathered at t2. The one week time lag was considered sufficient for participants to retain interest in the study, but not too long as they might be susceptible of losing interest [49]. Past research showed that MTurk is a platform that allows for collecting reliable and valid data in samples of gamblers [50,51]. The constructs and items used in the survey are shown in S1 Table in S1 File.

Mobile gambling apps were defined for the respondents as "*. . .bespoke apps for mobile devices, such as the Bet365 app or the Paddy Power app, and websites optimized for mobile gambling.*" Data were only collected from respondents who answered affirmatively to the initial filter question asking if they currently use mobile gambling apps. Of the total 525 initial responses, 86 participants did not complete both surveys, resulting in unusable data that was discarded[1] (ANOVA tests confirmed there were no significant differences between the 86 non-respondents and the main sample in their responses to survey 1). Another 112 responses were discarded as participants incorrectly answered one or both of the attention check questions included in the surveys (e.g. Canada is a country in Asia). After this data cleaning process, the final sample consisted of 327 participants. Given the number of independent variables in our model (N = 8), a minimum sample size of 262 was required to have valid predictive power at the 95% power level [52]. Accordingly, our study was sufficiently powered.

Of the 327 respondents, the majority were male (n = 230, 70%), lived in the United States (n = 173, 53%), with a mean age of 33 (SD = 8). Most participants had completed a bachelor degree education (62%). Concerning their mobile gambling activities, most gambled less or equal to three days a week (n = 218; 67%), had been using mobile gambling apps for 1–5 years (n = 158; 48%). The median amount gambled was US$50 per week with a minimum and maximum of US$5 and US$2,500 respectively. The most popular gambling apps used were Bet 365, PaddyPower, DraftKings, FOX Bet Sportsbook, and 888 Sports. 48% of respondents reported a score greater than 7 on the Problematic Gambling Severity Index (PGSI) which is indicative of problematic gambling [53]. The exact breakdown of the sample relative to age, gender and location is detailed in Table 1. Together, both surveys combined took an average of 11 minutes to complete with each of the 327 respondents receiving a payment of $2.50 (this equates a wage of approximately €13.50 per hour), which is consistent with ethical guidelines regarding participant's compensation in MTurk studies [54].

### Ethical approval

The study received ethical approval from the NUI Galway Research Ethics Committee (Ref: 18-Oct-10). All participants gave written consent to participate in this study.

### Measures

Except where specified, a seven-point Likert scale was used to measure the key constructs of the research model. All measurement items were taken from prior research and briefly

**Table 1. Respondent demographics.**

| Age | | Education | | Gender | | Gambling preference | |
|---|---|---|---|---|---|---|---|
| 18–20 | 2 | Less than high school | 4 | Male | 230 | Smartphone app | 230 |
| 21–25 | 22 | High School | 33 | Female | 95 | Laptop/computer | 76 |
| 26–30 | 108 | Some college but no degree | 35 | Non-Binary | 2 | Land bookmaker | 20 |
| 31–35 | 83 | Bachelor degree | 204 | | | Land casino | 1 |
| 36–40 | 44 | Postgraduate degree | 49 | | | | |
| 41–50 | 35 | PhD | 2 | | | | |
| 51–60 | 19 | | | | | | |
| >60 | 14 | | | | | | |

| Mobile Gambling Frequency over past week | | Location | | Gambling Experience | |
|---|---|---|---|---|---|
| One day | 61 | USA | 173 | <1 year | 11 |
| Two days | 84 | UK | 69 | 1–5 years | 155 |
| Three days | 73 | Australia | 26 | 5–10 years | 99 |
| Four days | 42 | India | 22 | 10–15 years | 30 |
| Five days | 31 | Ireland | 17 | 15–20 years | 18 |
| Six days | 21 | Canada | 11 | >20 years | 14 |
| Everyday | 15 | Other | 9 | | |

summarized here. All data and items used in the online survey can be obtained from https://osf.io/643ja/.

**Demographic characteristics.** Participants were asked to provide information about their age, gender, nationality, and educational attainment. Data on gambling preferences, mobile gambling frequency, gambling experience, and amounts gambled (i.e. amount staked in a typical week) were also sought and included as controls in the research model.

**Uses and gratifications.** The U&G scales for passing time (e.g. I use mobile gambling apps to occupy my time), entertainment (e.g. I use mobile gambling apps because it's entertaining), and information seeking (e.g. I use mobile gambling apps because it's a new way to do research for bets) were adapted from an existing study [33], as were the scales for arousal (e.g. I use mobile gambling apps because it raises my level of adrenaline; [26], relaxation (e.g. Using gambling apps allow me to unwind; [55], and social interaction (e.g. My friends and I use gambling apps as a reason to get together; [56]).

**Harmonious and obsessive passion.** The ten-item Gambling Passion Scale [6] was used to measure both harmonious (e.g. This gambling activity reflects the qualities I like about myself) and obsessive passion (e.g. I cannot live without this gambling activity) for mobile gambling.

**Problem gambling severity index (PGSI) scale.** The commonly used nine-item Canadian Problem Gambling Severity Index (PGSI) was used to assess disordered gambling symptoms [10]. For each question (e.g. Have you gone back on another day to try to win back the money you lost?), the individual was asked to answer "Never" (0), "Sometimes" (1), "Most of the time" (2), "Almost always" (3). The scores for each question were added together to create a total score of each individual.

**General mood.** The fourteen-item version of the Emmons Mood Indicator [57] was used to measure general mood. Participants were asked to reflect on the past month of their lives and rate fourteen emotion adjectives in terms of how frequently they experienced each using a seven-point Likert scale that ranged from 1 = "not at all" to 7 = "extremely much". Responses to negatively worded adjectives (e.g. angry, frustrated) were reverse scored and averaged with values from positively worded adjectives (e.g. happy, joy) to compute general mood scores for each participant.

## Results

The analysis was conducted using SmartPLS Version 3.3.2 [58]. The partial least squares (PLS) approach to structural equation modelling (SEM) enables researchers to estimate complex models with many constructs, indicator variables and structural paths without constraining distributional assumptions on the data [59]. In SEM analyses, a two-step approach, first examining the measurement model and then the structural model is recommended [60].

### Measurement model

The measurement model assesses convergent and discriminant validity. Convergent validity tests whether the items of a construct that are expected to be related are highly correlated. Good reliability is demonstrated if Cronbach's alpha (CA) is higher than 0.7 [61]. Composite reliability (CR) being above 0.8 and average variance extracted (AVE) values exceeding 0.5 are also suggested as evidence of sufficient convergent validity [62]. Table 2 shows the factor loadings, CA, CR, and AVE for each variable in the research model. Regarding CA, all constructs had a value higher than 0.7. All measures exceeded the minimum levels, with CR ranging from 0.88 to 0.96, while AVE ranged from 0.68 to 0.92, indicating good convergent validity.

Discriminant validity refers to whether the items measure the construct in question or other (related) constructs [63]. We evaluated the discriminant validity by comparing the square roots of AVE values to the inter-construct correlations [62]. The correlation matrix is shown in Table 3. As can be seen from the table, the square roots of the AVE values for the variables are consistently greater than the off-diagonal correlation values, suggesting satisfactory discriminant validity between the variables.

We evaluated the structural model by using the coefficient of determination and the significance level of each path coefficient [64]. The significance of path coefficients was determined via a bootstrapping procedure by setting the number of cases equal to the sample size and the number of bootstrap samples to 5,000. The results of the model are displayed in Fig 2.

Regarding the relationships between U&G and passion for mobile gambling, arousal is positively related to both harmonious ($\beta$ = .20, $p < 0.001$) and obsessive passion ($\beta$ = .20, $p < 0.001$). Seeking entertainment was negatively related to obsessive passion for mobile gambling ($\beta$ = -.22, $p < 0.001$) but unrelated to a harmonious passion. Information seeking on gambling bets was positively related to an obsessive passion ($\beta$ = .25, $p < 0.001$), and more strongly correlated to a harmonious passion for mobile gambling ($\beta$ = .31, $p < 0.001$). The relationship between passing time and obsessive passion was significant but weak ($\beta$ = .12, $p < 0.001$), and insignificant for a harmonious passion for mobile gambling. Relaxation was a strong predictor of harmonious passion for mobile gambling ($\beta$ = .39, $p < 0.001$) and moderately strong for obsessive passion ($\beta$ = .21, $p < 0.001$). Interestingly, social interaction was only significantly related to an obsessive passion for mobile gambling ($\beta$ = .29, $p < 0.001$). As social interaction was measured using a two-item construct, there is a risk this finding could be spurious. To ensure this was not the case, we retested the model using mean centred values for social interaction. The results of this approach matched the analysis reported. In terms of general mood, a harmonious passion for mobile gambling demonstrates a positive and strong relationship ($\beta$ = .40, $p < 0.001$), while obsessive passion is equally strong but inversely related ($\beta$ = -.39, $p < 0.001$). As would be expected, PGSI is very strongly correlated with an obsessive passion for mobile gambling ($\beta$ = .73, $p < 0.001$). However, the relationship between a harmonious passion for mobile gambling and PGSI is unsupported.

With respect to $R^2$, a variance of 52.3% is explained for harmonious passion for mobile gambling, and 47.1% is explained for obsessive passion; 45.2% of the PGSI variance and 15.3% for general mood is explained in our research model. Of the control variables, age was

**Table 2. Reliability, validity, and descriptives of the measurement model.**

| Variable | Item | *Mean* | SD | *K* | *Sk* | CA | Loading | CR | AVE |
|---|---|---|---|---|---|---|---|---|---|
| **Arousal** | ArL1 | 4.52 | 1.76 | -0.66 | -0.50 | 0.91 | 0.88 | 0.94 | 0.79 |
| | ArL2 | 4.61 | 1.79 | -0.68 | -0.53 | | 0.90 | | |
| | ArL3 | 4.81 | 1.67 | -0.19 | -0.69 | | 0.92 | | |
| | ArL4 | 5.19 | 1.50 | 0.66 | -0.98 | | 0.85 | | |
| **Entertainment** | Ent1 | 5.53 | 1.40 | 1.28 | -1.18 | 0.80 | 0.80 | 0.89 | 0.72 |
| | Ent2 | 5.11 | 1.53 | 0.13 | -.081 | | 0.86 | | |
| | Ent3 | 5.62 | 1.32 | 1.32 | -1.16 | | 0.89 | | |
| **Information Seeking** | Info1 | 4.08 | 1.88 | -1.17 | -0.24 | 0.91 | 0.89 | 0.94 | 0.80 |
| | Info2 | 4.13 | 1.95 | -1.14 | -0.24 | | 0.92 | | |
| | Info3 | 4.34 | 1.92 | -1.02 | -0.42 | | 0.92 | | |
| | Info4 | 4.60 | 1.85 | -0.70 | -0.63 | | 0.85 | | |
| **Pass Time** | PsTm1 | 4.57 | 1.75 | -0.78 | -0.46 | 0.86 | 0.87 | 0.88 | 0.79 |
| | PsTm2 | 4.22 | 1.86 | -1.09 | -0.17 | | 0.86 | | |
| | PsTm3 | 4.48 | 1.78 | -0.83 | -0.41 | | 0.91 | | |
| **Relaxation** | Relax1 | 4.61 | 1.70 | -0.49 | -0.55 | 0.87 | 0.83 | 0.89 | 0.79 |
| | Relax2 | 4.71 | 1.68 | -0.39 | -0.59 | | 0.92 | | |
| | Relax3 | 4.76 | 1.69 | -0.33 | -0.67 | | 0.91 | | |
| **Social Interaction** | Socl1 | 3.70 | 2.14 | -1.50 | 0.01 | 0.93 | 0.95 | 0.93 | 0.92 |
| | Socl2 | 3.68 | 2.16 | -1.42 | 0.11 | | 0.94 | | |
| **Harmonious Passion for Mobile Gambling** | HPass1 | 3.86 | 1.89 | -1.08 | -0.08 | 0.88 | 0.80 | 0.90 | 0.68 |
| | HPass2 | 4.19 | 1.79 | -0.94 | -0.21 | | 0.71 | | |
| | HPass3 | 4.03 | 1.82 | -0.93 | -0.18 | | 0.89 | | |
| | HPass4 | 3.88 | 1.80 | -1.05 | -0.04 | | 0.87 | | |
| | HPass5 | 4.31 | 1.75 | -0.75 | -0.38 | | 0.85 | | |
| **Obsessive Passion for Mobile Gambling** | OPass1 | 3.02 | 1.90 | -1.08 | 0.47 | 0.95 | 0.90 | 0.94 | 0.84 |
| | OPass2 | 3.28 | 2.02 | -1.30 | 0.32 | | 0.91 | | |
| | OPass3 | 3.36 | 2.07 | -1.35 | 0.25 | | 0.93 | | |
| | OPass4 | 3.34 | 2.08 | -1.30 | 0.31 | | 0.93 | | |
| | OPass5 | 3.16 | 2.03 | -1.17 | 0.44 | | 0.91 | | |
| **Problem Gambling Severity Index (PGSI)** | PGSI1 | 2.36 | 1.18 | -0.91 | 0.39 | 0.94 | 0.83 | 0.93 | 0.68 |
| | PGSI2 | 2.47 | 1.29 | -1.00 | 0.37 | | 0.84 | | |
| | PGSI3 | 2.81 | 1.24 | -1.03 | -0.02 | | 0.74 | | |
| | PGSI4 | 2.06 | 1.32 | -0.64 | 0.88 | | 0.84 | | |
| | PGSI5 | 2.25 | 1.29 | -0.84 | 0.62 | | 0.88 | | |
| | PGSI6 | 2.22 | 1.35 | -0.86 | 0.69 | | 0.81 | | |
| | PGSI7 | 2.44 | 1.32 | -1.03 | 0.49 | | 0.80 | | |
| | PGSI8 | 2.09 | 1.26 | -0.62 | 0.78 | | 0.84 | | |
| | PGSI9 | 2.18 | 1.36 | -0.77 | 0.77 | | 0.87 | | |
| **General Mood** | Mood1 | 5.53 | 1.28 | 0.06 | -0.78 | 0.96 | 0.88 | 0.96 | 0.81 |
| | Mood2 | 5.07 | 1.42 | -0.47 | -0.57 | | 0.90 | | |
| | Mood3 | 5.12 | 1.40 | -0.03 | -0.69 | | 0.92 | | |
| | Mood4 | 5.29 | 1.37 | -0.29 | -0.70 | | 0.92 | | |
| | Mood5 | 5.11 | 1.32 | -0.10 | -0.67 | | 0.89 | | |
| | Mood6 | 5.06 | 1.37 | -0.40 | -0.55 | | 0.91 | | |
| | Mood7 | 5.13 | 1.44 | -0.51 | -0.57 | | 0.88 | | |

Note: M = mean; SD = standard deviation; K = kurtosis; Sk = skewness.

**Table 3. Correlations between latent variables (square root of AVEs bolded in the main diagonal).**

| | 1 | 2 | 3 | 4 | 5 | 6 | 7 | 8 | 9 | 10 | 11 | 12 | 13 | 14 | 15 |
|---|---|---|---|---|---|---|---|---|---|---|---|---|---|---|---|
| 1.Age | **1.00** | | | | | | | | | | | | | | |
| 2.Amount | 0.06 | **1.00** | | | | | | | | | | | | | |
| 3.Arousal | 0.06 | -0.03 | **0.89** | | | | | | | | | | | | |
| 4.Education | 0.05 | 0.05 | 0.03 | **1.00** | | | | | | | | | | | |
| 5.Entertain | -0.02 | -0.12 | 0.54 | 0.02 | **0.85** | | | | | | | | | | |
| 6.Experience | -0.48 | -0.01 | -0.04 | -0.06 | 0.00 | **1.00** | | | | | | | | | |
| 7.Gender | -0.07 | -0.02 | -0.05 | 0.03 | -0.01 | -0.12 | **1.00** | | | | | | | | |
| 8.Mood | -0.16 | -0.03 | 0.06 | 0.08 | 0.28 | -0.01 | -0.07 | **1.00** | | | | | | | |
| 9.H Passion | 0.07 | 0.07 | 0.51 | 0.21 | 0.38 | 0.01 | -0.01 | 0.14 | **0.83** | | | | | | |
| 10.Info | 0.07 | 0.01 | 0.41 | 0.19 | 0.23 | 0.01 | -0.12 | 0.12 | 0.53 | **0.89** | | | | | |
| 11.O Passion | 0.19 | 0.14 | 0.45 | 0.20 | 0.22 | -0.08 | -0.05 | -0.12 | 0.67 | 0.50 | **0.92** | | | | |
| 12.PGSI | 0.15 | 0.14 | 0.33 | 0.07 | 0.06 | -0.09 | 0.00 | -0.24 | 0.39 | 0.40 | 0.66 | **1.00** | | | |
| 13. Pass Time | 0.16 | 0.02 | 0.38 | 0.07 | 0.48 | -0.11 | 0.09 | 0.02 | 0.31 | 0.06 | 0.31 | 0.14 | **0.89** | | |
| 14.Relax | 0.05 | -0.06 | 0.45 | 0.06 | 0.60 | -0.02 | 0.06 | 0.12 | 0.58 | 0.27 | 0.44 | 0.21 | 0.48 | **0.89** | |
| 15.Socal | 0.18 | 0.11 | 0.41 | 0.23 | 0.28 | -0.14 | -0.07 | 0.11 | 0.52 | 0.54 | 0.59 | 0.41 | 0.33 | 0.47 | **0.97** |

negatively related to general mood, amount gambled was positively related to PGSI, while gambling experience was negatively related to PGSI.

In terms of a goodness-of-fit measure for PLS-SEM, it is widely accepted that the standardized root mean square residual (SRMR) being less than 0.08 indicates a good fit [60]. The SRMR value for the for the saturated model was 0.04 and 0.06 for the estimated model, both of which are below the recommended cut-off value. Calculating the $Q^2$ is another mean to assess the predictive power of the PLS path model [65]. This measure indicates the model's out-of-sample predictive power and relevance. In a structural model, $Q^2$ values larger than zero for reflective latent constructs indicate predictive relevancy for a particular dependent construct [66]. In our model, all four dependent variables are above zero, with PGSI the highest $Q^2$

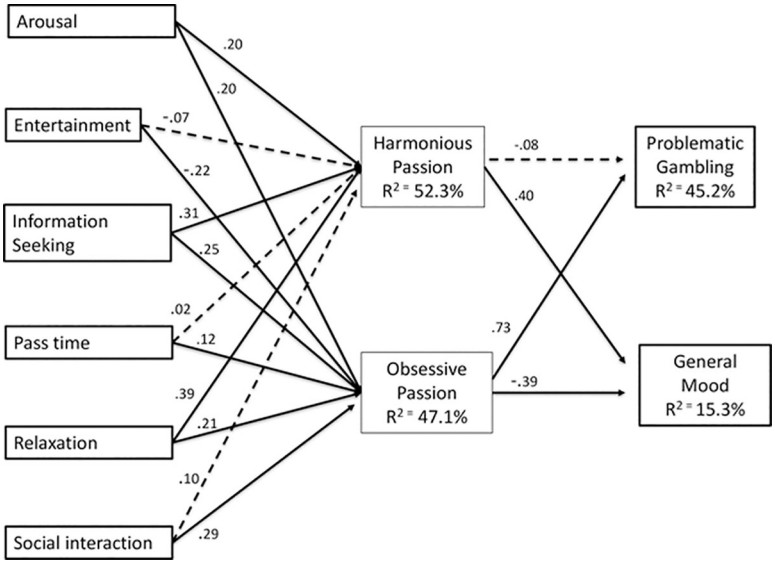

**Fig 2. Summary of the results.** Solid lines correspond to statistically significant coefficients, $p < .05$. Dashed lines correspond to non-significant coefficients, $p > .05$.

(0.43), followed by obsessive passion (0.39), harmonious passion (0.37), and deficient general mood (0.13). Thus, these results provide support for predictive relevancy of the latent constructs.

In a final assessment of the model's predictive validity, we conducted a PLSpredict analysis which uses a subsample of the data to predict another sample in a random repeated process. Researchers are advised to focus on the prediction errors of the model's key endogenous construct when interpreting PLSpredict results [66]. If none of the prediction errors for each construct indicator are higher for the PLS-SEM assessment when compared to a linear regression model, then a high predictive power can be assumed [66]. In our PLSpredict assessment, we selected obsessive passion for mobile gambling as the central construct. The linear regression prediction errors for all indicators were higher than the PLS-SEM version, thus confirming the model exhibits high predictive power. In conclusion, the data fits the measurement model.

## Discussion

The aim of this study was to address recent calls to disentangle problematic versus non-problematic intensive involvement in mobile gambling [7]. Capitalizing on previous work from having used the DMP framework in the video gaming field [67,68], and also relying on the U&G perspective of media consumption, we tested a model that aimed to segregate the antecedents and consequences of harmonious and obsessive passions for mobile gambling.

Our results show that the relationship between the U&G motives for engaging in mobile gambling and the outcomes of mood and problematic gambling are accounted for by the type of passion held for mobile gambling. Consistent with prior DMP studies conducted on various types screen-based activities [69–71], obsessive passion is associated to negative mood and gambling problems, while harmonious passion is associated with positive mood and unrelated to problematic gambling. Such findings are important from a public health and societal perspective, as mobile gambling is on the rise and the risk of over-diagnosis and pathalogization exist. The results of the current study thus suggest that similarly to what was found in relation to other digital technologies [67,68,72,73], high involvement in mobile gambling is not essentially problematic. Crucially, harmonious passion for mobile gambling is unrelated to disordered gambling symptoms, implying that such passion for gambling cannot be considered as a risk factor toward the development of disordered gambling habits (or as an initial step toward the development of problematic gambling patterns).

A unique contribution of this study is the application of the U&G lens to reveal how different media consumption gratifications fortify harmonious or obsessive passion for mobile gambling. While studies have examined gambling passions and their intrinsic and extrinsic motivational triggers [18,74], U&G offers a different perspective in that the desires and needs sought from digital media are central, and media competes with other resources to provide these gratifications [25,32,33]. Underpinned by the U&G framing, our study findings echo the notion of separate antecedents of gambling passions [18]. While the gratifications of arousal, information seeking, and relaxation were positively correlated to both types of passion, differences were observed for entertainment, passing time, and social interaction.

Seeking entertainment was inversely related to obsessive passion for mobile gambling and unrelated to harmonious passion. This result may be explained by the fact that obsessive passion is more often related to negative reinforcement (i.e. feeling less bad) [43]. Likewise, 'fun' motives are not problematic and are unconnected to gambling disorders as assessed by the SOGS [19]. Engaging in mobile gambling to pass time was associated, albeit weakly, to obsessive passion, but unrelated to the harmonious variety. This result contrasts with findings from a previous study showing that passing time is a characteristic of the casual player and not

necessarily the problematic one [65]. However, this inconsistency might be accounted for by the fact that the current study conceptualized passing time in a negative way (e.g. I use mobile gambling apps because it passes time when bored), implying that its association with obsessive passion is somewhat expected. These findings echo with the view that obsessive passion is driven by negative reinforcement (e.g. relief from boredom) and promote rigid and compulsive engagement that go beyond free will and self-control [43,68].

Social motivations have been extensively investigated in the context of gambling behaviors. For example, influential models [4] posits that social motives can constitute reinforcement promoting excessive gambling patterns, but to a lesser extent than others motives such as coping motives. In contrast, the present study finds that engaging in mobile gambling to facilitate social interaction was an important predictor of obsessive passion. No association was found for harmonious passion. This finding diverges somewhat from previous studies which endorse the protective role of social motives in gambling [19] and other problematic screen based behaviors such as video games [75,76] and binge watching [77]. However, our social interaction findings do align with the one previous study of the motivations underpinning passion for land-based gambling [18]. The link between social gratifications and obsessive mobile gambling passion could be a result of the broader cultural normalization of mobile gambling. As documented in McGee's [10] qualitative investigation, mobile sports gambling has become more socially accepted than the stigmatized betting shop. Indeed, for many of the men interviewed, placing bets through smartphones while socializing with their peers is now a standard aspect of sports fandom. It would seem social gratifications do not protect the obsessively passionate mobile gambler, but could actually promote problematic involvement in mobile gambling. However, some mobile gambling activities such as sports betting are more likely to facilitate social interaction over others like mobile slots [78]. Future research should determine if the relationship between social gratification and problem gambling is stronger for some mobile gambling activities than others.

As the psychological underpinnings of mobile gambling are different to land-based and other forms of Internet gambling [12], some practical consequences can be derived from the present study. First, the findings also have clinical implications by suggesting interventions and preventive actions which promote harmonious over obsessive involvement in mobile gambling. Unlike harmonious passion, obsessive passion is associated with a reliance on social interactions afforded by mobile gambling. Highly involved mobile gamblers could thus be encouraged to seek social affiliation and gratification through leisure activities outside of gambling. Likewise, regulated and willful choice in the activity is a defining feature of harmonious passion. Therefore, approaches which aid mobile gamblers to control the urge to gamble when in a state of boredom may prove fruitful. Controlling such inclinations is difficult as the habitual use of smartphones is a staple of modern life. Gambling blocking software, such as Gamban and Betblocker, could be used to prevent access to all gambling apps in at-risk situations (e.g., boring contexts). However, future research is required to determine the effectiveness of such blocking software.

Second, our findings have implications for policy makers. While gambling advertisements are banned in some jurisdictions, in countries such as Ireland and the UK, loose regulation is evident. Advertisements promoting mobile gambling apps often depict people enjoying gambling together in familiar social settings, the family home or in a bar. As evident in our findings, the intended normalization of mobile gambling as a social activity is potentially hazardous. Regulators wishing to promote responsible gambling could restrict gambling app promotions from depictions and associations with social inclusion.

## Limitations

Our study comes with several limitations. Data for measuring the proposed structural model were collected from MTurk. While previous work has suggested it to be an acceptable platform for research data collection [79], including for gambling-related research [50,51], participants were self-selected and might not be representative of the population in any of the geographical areas or age groups [54]. Accordingly, our findings should only be generalized with care. Despite data being collected in two waves, the model is still a cross-sectional representation, inviting future work to focus on the longitudinal development of obsessive and harmonious passions for mobile gambling. While we found harmonious and obsessive passion to be associated with mood, this could possibly be due the generally-positive framing of prompts in the harmonious passion scale, and the generally-negative framing of prompts in the obsessive passion scale. However, to mitigate this potential bias, the current study employed a two-wave methodology that avoid priming effect on our dependent variables (i.e. mood and problematic gambling). Theory-wise, we operationalized six prominent U&Gs as antecedents of harmonious and obsessive passion. These particular U&Gs were adapted from studies of mobile gaming and may not be the most appropriate for mobile gambling. Future work could expand this analysis by developing U&G measures specific to the mobile gambling context. For example, U&Gs such as coping-escapism and nostalgia may be fruitful in explaining the outcomes of mobile gambling.

## Conclusions

As the ability to gamble through a mobile phone is a relatively new phenomenon, there is a need to disentangle the causes and consequences of mobile gambling in problematic versus non-problematic populations. The present study contributes to this line of enquiry by investigating how gratifications associated with mobile gambling map to harmonious and obsessive passion, and ultimately mood and problematic gambling outcomes. While the gratifications of arousal, information seeking, and relaxation were positively correlated to both types of passion, differences were observed for entertainment, passing time, and social interaction. Additionally, obsessive passion for mobile gambling is associated with poor mood and problematic gambling while harmonious passion is associated with positive mood but is unrelated to problematic gambling. Based on our findings, intervention strategies for obsessive mobile gamblers should aim to unlink the activity from being a social pass time.

## Supporting information

**S1 File. Questionnaire.**
(DOCX)

## Author Contributions

**Conceptualization:** Eoin Whelan.

**Formal analysis:** Eoin Whelan.

**Methodology:** Eoin Whelan, Samuli Laato, A. K. M. Najmul Islam, Joël Billieux.

**Supervision:** Eoin Whelan.

**Writing – original draft:** Eoin Whelan, Samuli Laato, A. K. M. Najmul Islam, Joël Billieux.

**Writing – review & editing:** Eoin Whelan, Samuli Laato, A. K. M. Najmul Islam, Joël Billieux.

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
