## [Decision Letter · Decision Letter 0]

14 Dec 2020

PONE-D-20-32173

A Casino in my Pocket: Exploring the Antecedents and Consequences of Mobile Gambling Passion

PLOS ONE

Dear Dr. Whelan

Thank you for submitting your manuscript to PLOS ONE. After careful consideration, we feel that it has merit but does not fully meet PLOS ONE’s publication criteria as it currently stands. Therefore, we invite you to submit a revised version of the manuscript that addresses the points raised during the review process.

We look forward to receiving your revised manuscript.

Kind regards,

Daniel S McGrath, Ph.D

Academic Editor

PLOS ONE

Journal Requirements:

Reviewers' comments:

Reviewer's Responses to Questions

**Comments to the Author**

1. Is the manuscript technically sound, and do the data support the conclusions?

Reviewer #1: Partly

Reviewer #2: Yes

2. Has the statistical analysis been performed appropriately and rigorously? 

Reviewer #1: Yes

Reviewer #2: Yes

3. Have the authors made all data underlying the findings in their manuscript fully available?

Reviewer #1: Yes

Reviewer #2: Yes

4. Is the manuscript presented in an intelligible fashion and written in standard English?

Reviewer #1: Yes

Reviewer #2: Yes

5. Review Comments to the Author

Reviewer #1:

I would first like to extend my thanks for the opportunity to read this thoughtful, timely, and interesting submission, entitled “A Casino in my Pocket: Exploring the Antecedents and Consequences of Mobile Gambling Passion”. In this manuscript, the authors attempt to examine some antecedents and consequences of two forms of passion defined as either harmonious or obsessive. Using a sophisticated structural equation model and a large sample of online participants, the authors report asymmetric relationships between passion forms, and both gambling problems and general mood. These findings are well-founded, and clearly merit peer-reviewed publication.

At the same time, I have several significant concerns about the manuscript in its current form. These chiefly revolve around the introduction and discussion sections, as well as the theoretical conception of the passion construct in the context of addictive and habit-forming digital products like gambling apps. I have provided my comments, subdivided by section, below. Thank you once again for the opportunity to read this meritorious submission.

Abstract

1. The authors state that “only a minority of users develop uncontrolled gambling habits associated to tangible negative outcomes.” This characterization suggests that diagnosed Gambling Disorders are the only real or ‘tangible’ instances of harm arising from gambling. Multiple bankruptcies, and enduring troubles with one’s credit rating are certainly tangible harms that can be done by excess gambling, but would not contribute to a diagnosis of Gambling Disorder. The recognition of a broader sphere of sub-clinical gambling-related harms is part of the emerging Public Health perspective on problem gambling (e.g. Wardle et al., 2019, BMJ). The authors should remove or rephrase this passage to accurately reflect the broader landscape of gambling problems.

2. The authors suggest in this passage (and throughout the manuscript) that some patterns of highly-involved gambling are “adaptive” or “healthy”. I believe that this, too, is out of step with the available evidence, and beyond the scope of the authors’ obtained results. Large-scale longitudinal studies have clearly linked gambling involvement to gambling problems (Binde, Romild, & Volberg, 2017, IGS). The obtained results in this study do not show that harmonious passion is protective against gambling problems, but rather that it is not related to gambling problems (or the lack thereof). I believe that a more accurate characterization throughout this manuscript would say that harmonious passion could permit frequent gambling without elevating individuals’ existing risk of experiencing gambling problems.

3. The abstract appears to say there is only a small incidence of “tangible negative outcomes” associated with gambling involvement, but concludes with a suggestion to reduce harmful gambling by encouraging harmonious passion. Could the authors clarify the extent to which problem gambling is a serious concern in society?

4. The authors suggest that interventions focusing on obsessive passion “would be most effective in reducing problematic usage patterns.” I am not sure that this is implied by the evidence presented. One would need a comparative study involving multiple intervention types to reasonably suggest that any one of them could be “the most effective”. Would the authors be comfortable saying such an intervention “may be effective”, rather than “would be most effective”?

Introduction

1. For the reader’s convenience, on line 38, could the instructors stipulate roughly what percentage of the population experiences gambling problems (e.g. Potenza et al., 2019, Nature Reviews Disease Primers).

2. On line 45, could the authors provide a citation to support the claim that “some people gamble frequently with a sense of volition, and without experiencing negative consequences such as financial problems”. I am not aware of any studies that specifically look at volition during gambling activities.

3. On line 56, could the authors clarify what they mean by “sensors”? Coming from a slot machine research background, it is my view that modern slot machines can be highly sophisticated in many of the same ways as smartphones. Clarity is needed to understand what the authors mean when they say that smartphones afford unique gambling experiences.

4. On line 51, there may be a typo in the phrase “opportunities to gambling”.

5. On line 114, the authors compare gambling passion to sports, music, games, writing, work, and family. I wonder if it is appropriate to also compare gambling to other addictive substances and behaviours? Addictive things are unique in that they function to progressively diminish cognitive control, and so it seems reasonable to expect that passionate use of addictive things may naturally give way to uncontrolled, or ‘obsessive’ use patterns over time. Notably, this view seems to be contra-indicated by the authors’ non-significant correlations between gambling experience and both forms of passion. In any case, I am interested to know how the concept of passion could be applied to addictive things beyond gambling. Is it reasonable to say that there are passionate, high-frequency cocaine users, for whom drug use is both adaptive and healthy?

6. The authors’ definitions of harmonious and obsessive passion on line 122 stipulate that the key difference in harmonious passion is that the person “still remains in control” of their gambling behaviour. In some sense, it seems that the underlying construct of these forms of passion could be boiled down to whether or not gambling has become a habit, continuing irrespective of reward or punishment (e.g. Everitt & Robbins, 2016, Annual Review of Psychology). Reading this passage, I found myself frequently comparing these forms of passion to the constructs of “liking” versus “wanting” drugs (Robinson & Berridge, 1993, Brain Research Reviews). Could the authors clarify in this section the aspects of DMP that distinguish it from existing models of habit formation?

Methods

1. The methods stipulate that the PGSI was scored on a 4 point Likert scale from 0 to 3. However, the supplied OSF data repository shows that each PGSI item was scored on a 5-point Likert scale from 1 to 5. Did the authors use a non-standard scoring method for this instrument, or might there be some mistake in the data file? Does this impact the values reported in the tables? On a related note, readers would benefit from an accounting of the rate of problematic gambling (total PGSI score > 7) in the data set. The authors suggest several times that these results are applicable to problem gamblers, but that is not clear based on the available data, which only shows that there is significant positive skew on each of the nine PGSI items. To assess the applicability of these findings to problem gamblers, could the authors specifically state how many problem gamblers they tested?

2. The authors stipulate that participants used apps like Bet365 and PaddyPower, but that most participants were American. Could the authors clarify the extent to which participants used these specific apps, or provide some indication of which apps were being used by participants in the study?

3. The authors stated that the median participant gambled about $50 per week. Is that US dollars? Could they provide some indication of the range of expenditures that was reported in the sample (e.g. $5 – 1,500)?

4. The layout for Table 1 is quite difficult to follow. Different demographic questionnaires are confusingly intermingled. Could the authors separate these different questionnaires visually (e.g. by introducing white space between them)?

Results

1. The results are comprehensive and thoughtfully presented. The authors have done a great job here!

Discussion

1. The authors note here that obsessive passion is associated with negative mood, and harmonious passion with positive mood. Is it possible that these effects are driven by the generally-positive framing of prompts in the harmonious passion scale, and the generally-negative framing of prompts in the obsessive passion scale? For example, the HPass questions cover topics like appreciation (#3), liking oneself (#4), and having memorable experiences (#1), whereas the OPass questions cover topics like emotional dependence (#2), obsessions (#4), and being unable to live without gambling (#1). In essence, is it possible that people who generally feel good are more apt to endorse HPass, while people who feel bad are more apt to endorse OPass? If so, could this be reflected in the limitations?

2. On line 327, should “aim” read “aimed”?

3. The authors argue that “indicators such as frequency of gambling involvement should not be considered as [signs] of problematic gambling patterns” based on the obtained evidence. I do not believe that their findings negate the strong association between gambling involvement and gambling problems (Binde, Romild, & Volberg, 2017, IGS). Involvement may not – by itself – be a sufficient indicator of problem gambling, but it is clearly a valuable indicator of the likelihood that an individual may experience gambling problems: people who are uninvolved in gambling are less likely to experience gambling problems. I would like to recommend that the authors remove this passage.

4. The discussion section is quite lengthy, and some parts may not be absolutely germane to the authors’ exploratory analyses. In particular, the supplementary analysis in which social interaction reportedly mediates pass-time gratification in affecting Obsessive Passion is somewhat surprising to me, as it does not seem immediately obvious that mobile gamblers seeking a pass-time would do so seeking social interaction in mind. I have some further reservations about exploratory mediation modelling as an unreliable indicator of causality (e.g. Keele, 2015, American Journal of Evaluation). The authors have done a good job to keep from implying undue causality in this model, but I still wonder if it represents a necessary element of the manuscript.

Reviewer #2: Thank you for the opportunity to review the present manuscript that examined characteristics of obsessive and harmonious passion related to mobile gambling. Overall, I thought the manuscript was interesting and extremely well written and structured. I provide my comments below in hopes that it would help to further strengthen the authors manuscript.

1. The title is not really informative. Rather than antecedents and consequences of mobile gambling passion, perhaps a more accurate title would be something like characteristics associated with obsessive and harmonious passionate of mobile gambling?

2. The authors do a fantastic job providing a rationale for the present research. I think the introduction could be strengthened if the authors provide a summary of motives related to gambling. In particular, it would be helpful if the authors speak to previous research that potentially speaks to motives that may be associated with positive aspects of gambling, rather than gambling disorder. The authors may wish to also examine the Positive Play Index to further bring in previous research that looks at potential predictors of positive gambling experiences.

3. I am curious if there were any differences between the 86 participants who did not complete the follow-up and those who did. Did the authors consider using imputation rather than list-wise deletion, especially if there are significant differences in variables of interest in the two groups?

4. Please provide alphas from the present sample.

5. The authors spend a considerable time in the discussion regarding the social aspect of mobile gambling. Yet, is online gambling really social? I believe the authors do not have a measure of type of mobile gambling participants engaged in. For example, if a person was engaging in mobile slots, I would argue that is not really a social activity.

6. In limitations, I would not that UGs assessed were from mobile games. There are key differences in mobile games and mobile gambling, and thus some UGs for mobile gaming may not apply to mobile gambling.

6. PLOS authors have the option to publish the peer review history of their article (what does this mean?). If published, this will include your full peer review and any attached files.

Reviewer #1: **Yes: **W. Spencer Murch

Reviewer #2: No

---

## [Author Response · Author response to Decision Letter 0]

12 Jan 2021

We have provided a table which details how we have taken on board and responded to each of the reviewers comments.

---

## [Decision Letter · Decision Letter 1]

20 Jan 2021

A Casino in my Pocket: Gratifications Associated with Obsessive and Harmonious Passion for Mobile Gambling

PONE-D-20-32173R1

Dear Dr. Whelan

We’re pleased to inform you that your manuscript has been judged scientifically suitable for publication and will be formally accepted for publication once it meets all outstanding technical requirements.

Kind regards,

Daniel S McGrath, Ph.D

Academic Editor

PLOS ONE

Additional Editor Comments (optional):

Reviewers' comments:

Reviewer's Responses to Questions

**Comments to the Author**

1. If the authors have adequately addressed your comments raised in a previous round of review and you feel that this manuscript is now acceptable for publication, you may indicate that here to bypass the “Comments to the Author” section, enter your conflict of interest statement in the “Confidential to Editor” section, and submit your "Accept" recommendation.

Reviewer #1: All comments have been addressed

Reviewer #2: All comments have been addressed

2. Is the manuscript technically sound, and do the data support the conclusions?

Reviewer #1: Yes

Reviewer #2: Yes

3. Has the statistical analysis been performed appropriately and rigorously? 

Reviewer #1: Yes

Reviewer #2: Yes

4. Have the authors made all data underlying the findings in their manuscript fully available?

Reviewer #1: Yes

Reviewer #2: Yes

5. Is the manuscript presented in an intelligible fashion and written in standard English?

Reviewer #1: Yes

Reviewer #2: Yes

6. Review Comments to the Author

Reviewer #1: The authors have done a fantastic job addressing all of my original concerns. I have no further comments or recommendations, and I look forward seeing this valuable work in print.

Reviewer #2: The authors have addressed all my previous suggestions. I wish them the best of luck with their work!

7. PLOS authors have the option to publish the peer review history of their article (what does this mean?). If published, this will include your full peer review and any attached files.

Reviewer #1: No

Reviewer #2: No

---

## [Editor Report · Acceptance letter]

2 Feb 2021

PONE-D-20-32173R1 

A Casino in my Pocket: Gratifications Associated with Obsessive and Harmonious Passion for Mobile Gambling 

Dear Dr. Whelan:

I'm pleased to inform you that your manuscript has been deemed suitable for publication in PLOS ONE. Congratulations! Your manuscript is now with our production department. 

Kind regards, 

on behalf of

Dr. Daniel S McGrath 

Academic Editor

PLOS ONE